# Construct Validity and Responsiveness of Instruments Measuring Depression and Anxiety in Pregnancy: A Comparison of EPDS, HADS-A and CES-D

**DOI:** 10.3390/ijerph19137563

**Published:** 2022-06-21

**Authors:** Hanna Margaretha Heller, Stasja Draisma, Adriaan Honig

**Affiliations:** 1Department of Psychiatry, Amsterdam UMC Location Vrije Universiteit Amsterdam, Boelelaan 1117, 1081 HV Amsterdam, The Netherlands; 2Amsterdam Public Health, Mental Health Program, 1007 MB Amsterdam, The Netherlands; sdraisma@trimbos.nl (S.D.); a.honig@olvg.nl (A.H.); 3Department on Aging, Netherlands Institute of Health and Addiction (Trimbos Institute), Da Costakade 45, 3521 VS Utrecht, The Netherlands; 4Department of Psychiatry, OLVG Hospital, Jan Tooropstraat 164, 1061 AE Amsterdam, The Netherlands

**Keywords:** questionnaires, screening, structural validity, perinatal depression, perinatal anxiety, pregnancy, responsiveness

## Abstract

Depression and anxiety occur frequently in pregnancy and may have unfavourable consequences for mother and child. Therefore, adequate symptom measurement seems important. Commonly used instruments are the Center for Epidemiologic Studies Depression Scale (CES-D), the Edinburgh Postpartum Depression Scale (EPDS), and the Hospital Anxiety and Depression Scale, anxiety subscale (HADS-A). We compared the (1) structural and (2) longitudinal validity of these instruments. The data originated from a study on the effectiveness of an Internet intervention for pregnant women with affective symptoms. (1) A confirmatory factor analysis was used to estimate the construct validity. The theoretical factorial structure that was defined in earlier studies of the CES-D and the EPDS, but not the HADS-A, could be sufficiently replicated with acceptable CFI and RMSEA values. (2) Since there were two measurements in time, the hypotheses concerning plausible directions of the change scores of subscales that were (un)related to each other could be formulated and tested. In this way, longitudinal validity in the form of responsiveness was estimated. Ten of sixteen hypotheses were confirmed, corroborating the longitudinal validity of all constructs, except anhedonia, probably due to inconsistent conceptualization. The HADS-A seems less suitable to screen for anxiety in pregnancy. Anhedonia needs better conceptualisation to assess the change of symptoms over time with the CES-D and the EPDS.

## 1. Introduction

Depression and anxiety symptoms are regularly found among women during the antenatal period. Recent reviews show prevalence rates between 12 and 17 percent for antenatal depression [1,2] and prevalence rates of 15–23 percent of antenatal anxiety varying from pregnancy-specific anxiety to general anxiety symptoms [3,4,5], with higher rates in middle-low income countries [6,7]. Both antenatal depression and anxiety have been associated with poor pregnancy outcomes [8], poor postpartum mental health [9,10,11], and negative influences on child development [4,12,13,14,15]. Hence, screening for antenatal depression and anxiety is of the utmost importance. However, the selection of suitable screening instruments with adequate psychometric quality to identify symptoms of anxiety and depression during pregnancy is complicated: first, because physical changes, such as altered sleep and eating patterns, occur among most pregnant women, while items in self-report instruments for depression and anxiety often cover such somatic symptoms. This may inflate the estimated occurrence of anxiety and depression. Moreover, during pregnancy specific types of worries with respect to labour and the child often occur and are not addressed in commonly used self-report instruments. Therefore, they may be overlooked and underreported when assessed with these instruments [5].

For our study concerning the effectiveness of a guided Internet intervention [16] for pregnant women with depressive or anxiety symptoms, we chose three commonly used screening instruments that measure depressive and anxiety symptoms in pregnancy. These instruments were the Center for Epidemiologic Studies Depression Scale (CES-D) and the Edinburgh Postpartum Depression Scale (EPDS) for depressive symptoms, and the Hospital Anxiety and Depression Scale, anxiety subscale (HADS-A) for anxiety symptoms.

The CES-D is a self-rating scale meant for measuring depressive symptomatology in the general population, but it is also widely used in pregnancy [17,18]. The overview of Carleton et al. [19] of twenty-five studies investigating structural validity with factor analyses of this instrument found models with one to four factors. The authors proposed a new three factor structure, tested on five different samples.

The EPDS is the most common instrument that is used to assess depression, but it also includes a few anxiety items [20]. It was designed to assess postnatal depression, yet it has also been validated for antenatal depression [21]. However, its factor structure is not always well established or replicated, suggesting mostly three factors, including depression, anxiety, and anhedonia, but also two factors, including only depression and anxiety [22].

The HADS-A is part of the HADS, which consists of two reliable subscales, one for depression (HADS-D) and one for anxiety (HADS-A). These subscales can be used independently [23]. The HADS-A is a widely used anxiety screening scale, originally developed to assess anxiety in non-psychiatric hospital outpatients [24]. It does not contain questions about physical symptoms which makes it suitable for pregnant women [24,25] who experience physical changes that may relate to both anxiety as well as to pregnancy.

Various studies present psychometric properties of the CES-D, the EPDS and the HADS-A questionnaires [19,20,26,27,28], but few studies deal with the comparison of these psychometric properties in pregnancy [5,17,29,30]. To the best of our knowledge, only three studies were carried out in pregnancy and involved a comparison of the cross validity of the CES-D, the EPDS and the HADS-A based on single scores of these scales [17,30,31]. None assessed their longitudinal validity, which is defined by comparing their sensitivity to change. This is especially important in pregnancy due to the variability of depressive and anxiety scores [32,33] and for the assessment of treatment outcomes. Sensitivity to change is also referred to as the responsiveness of an instrument and can be defined as the ability of the instrument to detect clinically important change over time [34].

The first aim of our study is to assess the (structural) construct validity of the CES-D, the EPDS and the HADS-A in a population of pregnant Dutch women by using a confirmatory factor analysis (CFA) cross-sectionally with baseline data. Most studies about the structure of these three instruments only applied an exploratory factor analysis. However, a confirmatory factor analysis is more appropriate to replicate earlier found factor structures as part of the validation of an instrument.

Secondly, validity is explored in a longitudinal way by formulating and testing hypotheses about relations between the change scores for different subscales. We do this to estimate the responsiveness of instruments over time in pregnancy to explore their sensitivity in measuring changes of symptoms. The hypotheses concern the subthemes: depression, anxiety and anhedonia, and are presented in the methods section.

## 2. Methods

### 2.1. Design and Participants

A secondary data analysis was performed with data that were collected in a randomized controlled trial (RCT) concerning the effectiveness of a guided Internet intervention (MamaKits) for pregnant women with moderate to severe symptoms of anxiety or depression, or both. The MamaKits study is described in more detail elsewhere [16]. In short, the study included self-referred pregnant women expressing interest in an Internet intervention to treat their depressive or anxiety symptoms. The inclusion criteria were being aged above 18 years, less than 30 weeks pregnant, having depressive symptoms above threshold (i.e., CES-D > 16) or anxiety symptoms above threshold (i.e., HADS-A > 8), or both. The single exclusion criterion was being suicidal. All participants signed their informed consent. The participants were randomly allocated to receive either a guided Internet intervention or care as usual. The EPDS was used as an additional measurement instrument. Further information was collected using one question of the Web Screening Questionnaire (WSQ) to asses suicidality [35], the Trimbos/institute for Medical Technology Assessment, Erasmus University Rotterdam questionnaire for costs associated with Psychiatric Illness for additional information about mental health [36], de Client Satisfaction Questionnaire (CSQ-8) to measure satisfaction about the intervention [37] and data on perinatal child outcome through self-report. Eligible women were recruited throughout the Netherlands via general media and flyers in prenatal care waiting rooms or via obstetricians and midwives. A total of 349 women expressed interest in the study, of which 91 withdrew their participation for several reasons, such as feeling better already. A further 99 were excluded because they did not fulfil the inclusion criteria, such as not reaching the symptom threshold. A total of 159 women completed one measurement at inclusion (T0) and met the inclusion criteria. About 10 weeks after randomization 119 of them also filled out the second measurement (T1). Due to inclusion in different stages of pregnancy T0 and T1 were taken at different moments in pregnancy. The study received approval from the Medical Ethics Committee of the VU University Medical Center (2013.275) and was registered in the Netherlands Trial Register (NL4162). All data were collected by Internet.

### 2.2. Measures

#### 2.2.1. CES-D

The Center for Epidemiologic Studies Depression Scale (CES-D) [18] is a widely used instrument for screening depressive symptoms, also used in pregnancy, with 20 items concerning the last seven days. Symptoms are scored on a Likert-type scale from 0 to 3 (“rarely or none of the time” to “most or all of the time”), with a total score range from 0 (no depressive symptoms) to 60 (high number of depressive symptoms). The standard cut-off score is ≥16 for possible depression. The validity of the CES-D has been investigated in different populations, including pregnant women [30,38,39], on paper and online [40]. The scale has a sensitivity of 95.1% and a specificity of 85.0% [39] and the internal consistency (Cronbach’s alpha) of the online version in the population at large is 0.89–0.93 [40]. The factor structure varies but the most frequently used consists of the following three or four factors: Somatic symptoms (6 items: item 1, 2, 5, 7, 11 and 20); negative affect (depressed) (4 items: item 3, 6, 14 and 18); positive affect (anhedonia) (4 items: item 4, 8, 12 and 16); and interpersonal affect if four factors were used (two items: item 15 and 19) (Carleton 2013). However, in other studies, a two, or another three or four factor model was found [40,41,42]. Anxiety is not included as a factor, but the instrument as a whole contains one item about anxiety (item 10, “I felt fearful”).

#### 2.2.2. EPDS

The EPDS is a 10-item depression scale, primarily developed to detect depression in the postpartum period, but it is also validated and widely used during pregnancy [21]. It is also used to measure anxiety [5,22,43] and its items address depressive and anxiety symptoms concerning the past seven days. Depending on the trimester, the cut-off score varies worldwide from 6.5 to 14.5, and in the Netherlands, it varies from 10 to 11 [21]. The item response options are 0 to 3, and the total score range is 0 to 30 [21]. The internal consistency (Cronbach’s alpha) was 0.87–0.90 in an online version [40,44]. Several factor models are investigated, but in most studies a three-factor model seems to fit best to the data [20,45]. The three factors most frequently found are: anhedonia (2 items: 1 and 2), anxiety (4 items: 3, 4, 5 and 6) and depression (4 items: 7, 8, 9 and 10) [5,20,45,46,47].

#### 2.2.3. HADS-A

The Hospital Anxiety and Depression Scale, anxiety subscale (HADS-A) [24,27,48] is a screener for anxiety with item responses on a 0 to 3 scale, concerning the last week. Internal consistency as calculated with Cronbach’s alpha varies but is 0.89 in the paper version [48] and 0.80 in the Internet version [49]. The HADS-A consists of 7 items (item 1, 3, 5, 7, 9, 11, 13) and has an optimal cut-off ≥8 to predict an anxiety disorder, with a sensitivity of 0.89 and a specificity of 0.75 [24,25,48]. The total score range is 0 to 21. The HADS-A is also used to measure anxiety in pregnancy [25,32] and it consists of one factor.

### 2.3. Data Analysis

#### 2.3.1. Construct Validity

A confirmatory factor analysis (CFA) is commonly used to evaluate the latent factor structure of instrument items, to find support for an assumed grouping of items into subscales. For conceptualization of the factors of the three instruments that were involved, the factor structure was restricted a priori, according to the subscales that were found in recent studies. According to the study of Carleton [19], which consisted of five medium to large samples of people with different backgrounds, ages, gender, with and without medical and psychiatric problems, a three factor structure was found for the CES-D, consisting of the factors somatic symptoms, negative affect and anhedonia. The three-factor structure that was evaluated for the EPDS consisted of anhedonia, anxiety and depression [20,45,46,47]. The original HADS contains two factors, HADS-A- anxiety- and HADS-D –depression [26]. Since the MamaKits study only applied the HADS-A part of the instrument, only this factor, anxiety, was calculated and evaluated.

The LAVAAN package in R was used to assess whether the factors that were identified in the literature could be reproduced. Data of 159 included women from the baseline assessment were used. As measures for model fit, RMSEA (root mean square error of approximation) and CFI (comparative fit index) were used. RMSEA was assumed to be close to good when it was between 0.05 and 0.08, worse when >0.08 [50], TLI cut-off value close to 0.95 and SRMR cut-off value to 0.8. The CFI was acceptable when ≥0.90 and good when ≥0.95 [51].

#### 2.3.2. Responsiveness

We studied the responsiveness of the three instruments by formulating hypotheses about expected differences and similarities in the change scores between T0 and T1. The hypotheses were based on theoretical contrasts between the different factors. An example is that the change scores between anxiety factors of different instruments are presumed to correlate more with each other than a change score of an anxiety factor with a depression factor. A hypothesis was corroborated when the expected difference in correlations of the change scores was at least 0.1 [52]. We used the criteria of de Boer [53] to assess responsiveness. This states that responsiveness is high if less than 25% of the hypotheses are refuted, moderate if 25 to 50% are refuted and low if more than 50% are refuted. The factors that we studied were ‘negative or depressive affect’, ‘anxiety’ and ‘anhedonia’. We did not use the ‘somatic symptoms’ factor of the CES-D because this factor does not feature in any of the other instruments. The factor ‘negative or depressive affect’ refers to depressed mood (criterion A1 DSM-5 depressive disorder) [54]. The second factor, ‘anhedonia’ (criterion A2 DSM-5 depressive disorder) refers to the absence of almost all positive feelings and the inability to enjoy most or all aspects of life. The third factor ‘anxiety’ forms the main criterion of all anxiety disorders, as well as an additional feature to the classification of DSM-5 depressive disorder.

We formulated the following hypotheses:

Negative (depressive) affect We expected that the change scores in different subscales measuring depressive (negative) affect would be more strongly correlated to each other than to anxiety subscales (hypothesis 1 and 2) because depressive affect subscales contain no items measuring anxiety. Second, we hypothesized that the change scores of CES-D negative affect would correlate more with the change scores in EPDS depression than with the EPDS anhedonia change scores (hypothesis 3), because negative/depressive affect scales refer to gloominess or a black mood, whereas anhedonia reflects the in ability to enjoy life.

Anhedonia We expected that EPDS and CES-D scales measuring change in anhedonia scores would correlate more with each other than with the EPDS depression, the CES-D negative affect and the HADS-A anxiety scales (hypotheses 4, 5, 6) since they measure the same construct expressing the absence of the ability to enjoy life, which is conceptually different from anxiety and negative/depressive affect.

Anxiety First, we expected the CES-D change in item 10 anxiety to be more strongly correlated with the HADS-A total scale change, since they indicate the same construct, whereas the CES-D negative affect, the EPDS depression, the CES-D anhedonia and the EPDS anhedonia represent other factors (hypotheses 7, 8, 9 and 10). Second, we hypothesized that the change score of the HADS-A and the change score of the anxiety scale of the EPDS correlate more with each other than with those of the CES-D negative affect, the EPDS depression, the EPDS anhedonia and the CES-D anhedonia (hypotheses 11, 12, 13 and 14). Finally, we hypothesized that the HADS-A change score is equally correlated to the EPDS anxiety score as to the CES-D anxiety score (hypothesis 15).

Comparison of questionnaires We expected a stronger correlation of changes between the CES-D and the EPDS total scales than each instrument with the HADS-A (hypothesis 16) since the EPDS and the CES-D consist of almost the same factors, while the HADS-A contains only one common (anxiety) factor.

## 3. Results

### 3.1. General Results

A total of 159 women who completed T0 and met the inclusion criteria were enrolled, of which 74.8% (*n* = 119) completed T1. Their median CES-D scores at the two time points (T0, T1) were 28 and 17 (interquartile range were 9–48 and 2–55, respectively); the median EPDS scores were 14 and 8 (interquartile range, respectively, 3–28 and 0–26); and the median HADS-A scores were 12 and 8 (interquartile range, respectively, 4–20 and 1–19). A more extensive description of the results is described elsewhere [16]. Table 1 contains the description of the sample.

### 3.2. Cross-Sectional Results

Cronbach’s α’s for the total scales of the three instruments were: CESD 0.84, EPDS 0.80 and HADS-A 0.72, denoting the sufficient reliability of the scales. Results of the CFA (Table 2) indicated that the theoretical factorial structure of both EPDS [20] and CESD [19] could be replicated sufficiently with acceptable CFI’s and cut-off values of RMSEA’s that were close to 0.08. However, the one factor model for the HADS-A was not replicated adequately; RMSEA was higher than the cut-off of 0.08 and CFI was too small. The SRMR values were close to 0.07. The acceptable range for the SRMR index is between 0 and 0.08 [51], therefore these values are adequate. The TLI values are all rather low, lower than the required 0.95. These values indicate poor model fit, maybe due to low inter-item correlations. A notable finding in Table 3 is that two items about restlessness did not load on the same factors as in the previous research, neither in the factor somatic symptoms of the CES-D (item 11) [19], nor in the factor anxiety of the HADS (item 11) [26]. Both items correlated weakly with other items of the same factor.

### 3.3. Responsiveness Results

The majority of hypotheses–10 out of 16–concerning change could be confirmed, 37.5% were rejected, which indicates moderate responsiveness (Table 4).

More specifically, the change scores in CES-D and EPDS measuring depressive (negative) affect were more correlated to each other than to the changes scores in anxiety, but unexpectedly, they were also correlated to the EPDS factor anhedonia (difference 0.064).

Change scores of the factors measuring anhedonia were even less correlated to each other than to the factors measuring negative affect/depression and almost equally to the HADS-A.

The change scores of the anxiety items of all instruments were equally strongly correlated. However, the correlation of the change scores between the anxiety item (s) of the CES-D and the HADS-A was almost equal to the correlation with the EPDS anhedonia change (difference 0.04). This also applied to the correlation of change between the anxiety items of EPDS and the HADS-A with the CES-D anhedonia change (difference 0.08).

Concerning the measurement instruments as a whole, the change scores of the EPDS correlated more strongly with the CES-D change scores than with the HADS-A change scores, as was expected.

## 4. Discussion

Our evaluation of cross-sectional construct validity as assessed by CFA of the three instruments for measuring depression and anxiety in pregnancy, delivered predominantly adequate results, apart from those of HADS-A.

For the CES-D, the three-factor structure (‘somatic symptoms’, ‘negative affect’ and ‘anxiety’ [19]) was replicated sufficiently. The EPDS three-factor structure of anhedonia, anxiety and depression [20] was also found to be adequate. The factor structure of the HADS-A turned out relatively weak according to low fit measures. One possible explanation for the poor outcomes of the total HADS-A subscale in our study is that only HADS-A anxiety items were used, whereas in a meta-CFA it was found that the HADS mainly measures distress, without a distinction between depression and anxiety [26,28]. Furthermore, the HADS-A seems to be sensitive to biological changes, assessing anxiety as autonomic arousal [28,55]. Since biology during pregnancy changes considerably, this could negatively impact the validity of the instrument. Unlike the HADS-A, the EPDS item formulations for anxiety contain more cognitive-emotional expressions (such as, ‘feeling scared for no good reason’, ‘being anxious or worried for no good reason’) which may explain our replication of the proposed latent structure of the EPDS. However, since this scale was especially designed for use in the perinatal period, items that could be influenced by physical symptoms were avoided [56]. A third reason for the poor outcome of the HADS could be that the anxiety in pregnancy is different from other anxiety disorders and specific for the prenatal period [57].

In our study we added a new approach by exploring construct validity in a longitudinal way. We did this by formulating and testing hypotheses to assess the responsiveness of instruments over time and their difference in measuring changes of symptoms. As the majority of hypotheses concerning change scores could be confirmed, six of eight hypotheses considering anhedonia (hypotheses 3, 4, 5, 6, 10 and 14) had to be refuted. The most remarkable finding was that the correlation of change scores of the EPDS construct anhedonia was even less strongly correlated to the change score of the CES-D construct anhedonia than to the change scores of the constructs of CES-D negative affect and EPDS depression. It was about equally strongly correlated to the HADS anxiety scale. An explanation for this finding is the inconsistent conceptualization of the concept anhedonia. In the DSM-5, anhedonia is defined as the absence of the ability to have interest and/or enjoy activities that were previously considered pleasurable [54]. However, in the description of the factors of the CES-D, for example by Carleton [19] anhedonia is defined as the absence of positive affect. Considering the EPDS validation studies [20,46] the factor anhedonia is sometimes defined as inability to feel pleasure from normally pleasurable experiences and in other studies as having low positive affect.

So, although the CES-D and the EPDS both contain a factor anhedonia, the items loading on the factors differ substantially, which make them not comparable.

## 5. Limitations

The instruments were used in different stages of pregnancy, reducing comparability and hence, validity. Second, including the HADS-D instrument for depression could have resulted in better outcomes for the factor analysis of the HADS. Third, to deliver more robust results of the CFA of the instruments, we probably needed to include a larger sample of women. This is especially applicable to the TLI and the RMSEA [51]. Fourth, the fact that the women in our sample were relatively highly educated may have influenced the results of the cross sectional, as well as the longitudinal analyses.

## 6. Conclusions

Based on their high construct validity, CES-D and the EPDS, but not the HADS-A, seem to be reliable instruments to assess depression and anxiety in pregnancy.

The reason that HADS-A is probably less useful as an instrument to measure prenatal anxiety is its sensitivity to biological changes which occur frequently in pregnancy. Secondly, prenatal anxiety seems to be a distinct kind of anxiety which requires a specific type of questionnaire.

The responsiveness of the three instruments was moderate, probably due to the change scores of the anhedonia constructs which need more theoretical and empirical substantiation.

More research is needed to develop a sensitive questionnaire to measure anxiety in pregnancy and to investigate how to improve the responsiveness of instruments. Existing questionnaires measuring pregnancy-specific anxiety, such as the Pregnancy Related Anxiety Questionnaire and the Pregnancy-Related Anxiety scale need more extensive validation.

Furthermore, we recommend testing of the instruments in larger samples of women at the same stage of pregnancy, but with a more diverse background, such as a varying level of education.

## Figures and Tables

**Table 1 ijerph-19-07563-t001:** Sociodemographic and clinical characteristics at baseline for intervention group and control group.

Variables	
Demographic factors (*n* = 159)	
Maternal age, years (mean ± SD)	32.01 (4.71)
Background (Dutch)	134 (84.3%)
Education ^a^	
low	4 (2.5%)
middle	35 (22.0%)
high	120 (75.5%)
Marital status	
Relationship, yes	152 (95.6%)
Living together	144 (90.6%)
Employed, yes	111 (69.8%)
Pregnancy (*n* = 159)	
Duration by study entrance	
<12 weeks	16 (10.1%)
>12 and <26 weeks	92 (57.9%)
>26 weeks	51 (32.1%)
Previous mental health ^b^ (*n* = 159)	
Depressive disorder	53 (33.3%)
Anxiety disorder	45 (28.3%)
Other mental problems	11 (6.9%)
No diagnosis	61 (38.4%)
Affective symptoms (mean ± SD)	
T0 (*n =* 159)	
CES-D	28.38 (8.31)
EPDS	14.11 (4.91)
HADS-A	11.67 (3.43)
T1 (*n =* 119)	
CES-D	19.02 (9.74)
EPDS	9.17 (5.52)
HADS-A	8.52 (3.91)

^a^ Dutch Standard Classification of Education: Standaard Onderwijsindeling 2006—Editie 2016/’2017, StatLine, the electronic database of Statistics Netherlands. ^b^ note that women can be both in the category ‘depressive disorder’ and in the category ‘anxiety disorder’.

**Table 2 ijerph-19-07563-t002:** Fit measures of confirmatory factor analysis of the three instruments (*n* = 159).

	Chi Square (*p* Value)	CFI	TLI	RMSEA (CI)	SRMR
CESD (3 factors somatic, neg. affect/depression, anhedonia)	118.848 (df = 74, *p* = 0.001)	0.923	0.906	0.062 (0.040–0.082)	0.070
EPDS (3 factors, anxiety, depression, anhedonia)	62.55 (df = 32, *p* = 0.001)	0.929	0.899	0.077 (0.048–0.106)	0.069
HADS (1 factor anxiety)	51.996 (df = 14, *p* = 0.000)	0.831	0.747	0.131 (0.093–0.171)	0.075

**Table 3 ijerph-19-07563-t003:** Factor loadings for items of the CESD, HADS-A and EPDS (*n* = 159).

Factor	Item Content		Symptoms	
	CES-D	Somatic	Depression	Anhedonia
Somatic	1. I was bothered by things that usually don’t bother me.	0.297		
	2. I did not feel like eating; my appetite was poor.	0.356		
	5. I had trouble keeping my mind on what I was doing.	0.300		
	7. I felt that everything I did was an effort.	0.583		
	11. My sleep was restless.	0.048		
Depressed	3. I felt that I could not shake off the blues.		0.581	
	6. I felt depressed.		0.669	
	14. I felt lonely.		0.476	
	18. I felt sad.		0.598	
	20. I could not get ‘‘going’’		0.557	
Anhedonia	4. I felt that I was just as good as other people.			0.368
	8. I felt hopeful about the future.			0.298
	12. I was happy.			0.533
	16. I enjoyed life.			0.593
	EPDS	Anhedonia	Anxiety	Depression
Anhedonia	1. I have been able to laugh and see the funny side of things.	0.509		
	2. I have looked forward with enjoyment to things.	0.766		
Anxiety	3. I have blamed myself unnecessarily when things went wrong.		0.392	
	4. I have been anxious or worried for no good reason.		0.521	
	5 I have felt scared or panicky for no very good reason.		0.553	
	6. Things have been getting on top of me.		0.385	
Depression	7. I have been so unhappy that I have had difficulty sleeping.			0.505
	8. I have felt sad or miserable.			0.606
	9. I have been so unhappy that I have been crying.			0.641
	10. The thought of harming myself has occurred to me.			0.437
	HADS-A	Anxiety		
Anxiety	1. I feel tense or wound up.	0.384		
	3. I get a sort of frightened feeling as if something awful is about to happen.	0.648		
	5. Worrying thoughts go through my mind.	0.511		
	7. I can sit at ease and feel relaxed.	0.376		
	9. I get a sort of frightened feeling like ‘butterflies’ in the stomach.	0.432		
	11. I feel restless as if I have to be on the move.	0.140		
	13. I get sudden feelings of panic.	0.413		

**Table 4 ijerph-19-07563-t004:** Outcome of hypotheses for responsiveness of constructs of CES-D, HADS-A and EPDS.

Hypothesis:	Correlations *	Confirmed
CES-D negative affect change is more strongly correlated to EPDS depression change		
1 than to HADS-A (anxiety) change	0.694 vs. 0.313	Yes
2 than to EPDS anxiety change	0.694 vs. 0.248	Yes
3 than to EPDS anhedonia change	0.694 vs. 0.630	No < 0.1
CES-D anhedonia change is more strongly correlated to EPDS anhedonia change		
4 than to EPDS depression change	0.486 vs. 0.543	No
5 than to CES-D negative affect change	0.486 vs. 0.510	No
6 than to HADS-A change	0.486 vs. 0.462	No < 0.1
CES-D anxiety change (1 item) is more strongly correlated to HADS-A change		
7 than to CES-D negative affect change	0.458 vs. 0.286	Yes
8 than to EPDS depression change	0.458 vs. 0.355	Yes
9 than to CES-D anhedonia change	0.458 vs. 0.345	Yes
10 than to EPDS anhedonia change	0.458 vs. 0.418	No < 0.1
HADS-A (anxiety) change is more strongly correlated to EPDS anxiety change		
11 than to CES-D negative affect change	0.554 vs. 0.313	Yes
12 than to EPDS depression change	0.554 vs. 0.418	Yes
13 than to EPDS anhedonia change	0.554 vs. 0.295	Yes
14 than to CES-D anhedonia change	0.554 vs. 0.462	No < 0.1
HADS-A anxiety change is equally correlated to EPDS anxiety change		
15 than to CES-D anxiety (1 item) change	0.554 vs. 0.458	Yes
CES-D total scale change score is more strongly correlated to EPDS total scale change score		
16 than to HADS-A change score	0.732 vs. 0.485	Yes

* Correlations are calculated between change scores (T0–T1) of two subscales. Resulting correlations of two sets are compared, differences > 1 are considered significant.

## Data Availability

Data sharing is not applicable to this article. No new data were created or analysed in this study.

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
