# Peer review of "Construct Validity and Responsiveness of Instruments Measuring Depression and Anxiety in Pregnancy: A Comparison of EPDS, HADS-A and CES-D"

_ijerph, 2022, doi:10.3390/ijerph19137563_

Round 1

Reviewer 1 Report

This work compares structural and longitudinal validity of three instruments for depression and anxiety symptom measurement for pregnant woman. Results show validity of constructs (except for anhedonia) and strenghten the use of these instruments for pregnancy. The paper is well written, and a worthy scientific soundness is provided. Some minor points could be revised:

  1. Introduction, methods and results are properly described. Nevertheless, for non-expert readers, it would be interesting to provide a rationale about the implications of the results on the real practice. What actions the results encourage to perform?
  2. Limitations of this work are valuable. How do they affect to conclusions? What future research actions should it be taken?
  3.  Revise format of table 3
  4. Line 34: %percent > percent

Author Response

This work compares structural and longitudinal validity of three instruments for depression and anxiety symptom measurement for pregnant woman. Results show validity of constructs (except for anhedonia) and strengthen the use of these instruments for pregnancy. The paper is well written, and a worthy scientific soundness is provided. Some minor points could be revised:

We thank the reviewer for the kind words and will answer the questions below.

  1. Introduction, methods and results are properly described. Nevertheless, for non-expert readers, it would be interesting to provide a rationale about the implications of the results on the real practice. What actions the results encourage to perform?

As we stated in our conclusion we will advise not to use the HADS to measure prenatal anxiety. However, we agree with the Reviewer that we did not make clear recommendations. Therefore, we added to the conclusion the following sentence: “More research is needed to develop a sensitive questionnaire to measure anxiety in pregnancy and to investigate how to improve responsiveness of instruments.”

  1. Limitations of this work are valuable. How do they affect to conclusions? What future research actions should it be taken?

We thank the Reviewer for making these recommendations . We now added to the conclusion the following sentence: “Furthermore, we recommend testing of instruments in larger samples of women at the same stage of pregnancy, but with a more diverse background, such as a varying level of education.”

  1.  Revise format of table 3 Thank you for calling our attention to this. We revised the format.
  2. Line 34: %percent > percent We changed  it and removed “%”

    We hope that we have addressed the comments made by the reviewer in such a way that our manuscript is now suitable for publication in IJERPH . We are looking forward to your decision and remain available to answer any further questions.

    Sincerely yours, on behalf of all co-authors,

    Hanna Heller,

Reviewer 2 Report

The article presents a very interesting subject, especially for those who research the relationship between depression and motherhood. The comparison between the instruments is valid to understand which instruments to use, why, and what to expect from each one of them.

The manuscript, however, is written confusingly, making it difficult for the reader to understand the contextualization of the problem and the design of the research method.

I send some observations to assist in the improvement of the article.

Abstract: it is not clear how the research was carried out. The method should be further described, even in a summarized way, due to the number of words;

Introduction:

The first paragraph is unclear. The prenatal, perinatal, and postnatal phases were addressed. They are different phases and each one presents its characteristics, causes and they are all together as if they were the same situation. What did the authors intend with this explanation? Which phase is the focus of the study?

Writing should be reviewed, especially about grammar and syntax.

What would "mama kits" be? The numbers of participants are different in terms of methods and results.

What were the inclusion and exclusion criteria?

I believe that this review is necessary for the reader to fully understand the findings and discussion.

Results:  This paragraph could be in the method (participants).

"A total of 349 women expressed interest in the study of which 91 women withdrew 214 their participation for several reasons like feeling better already and 99 were excluded 215 because of not fulfilling the inclusion criteria like not reaching the symptom threshold. A 216 total of 159 women, who completed T0 enrolled and 74.8 % (n=119) completed T1."

Author Response

Reviewer 2 artikel vragenlijsten (article questionnaires)

The article presents a very interesting subject, especially for those who research the relationship between depression and motherhood. The comparison between the instruments is valid to understand which instruments to use, why, and what to expect from each one of them.

The manuscript, however, is written confusingly, making it difficult for the reader to understand the contextualization of the problem and the design of the research method.

I send some observations to assist in the improvement of the article.

Abstract: it is not clear how the research was carried out. The method should be further described, even in a summarized way, due to the number of words;

We thank the Reviewer for showing interest and apologize that the abstract was not clear regarding the way our research was carried out . We hope that our revision add to the clarity of our research method.

Introduction:

The first paragraph is unclear. The prenatal, perinatal, and postnatal phases were addressed. They are different phases and each one presents its characteristics, causes and they are all together as if they were the same situation. What did the authors intend with this explanation? Which phase is the focus of the study?

We apologize to the Reviewer that we were not clear regarding the perinatal phase which our study is addresses. We made changes to clarify which perinatal phase is studied and hope this will make the text more comprehensible.

We started the introduction with: “Depression and anxiety are common problems in women during the antenatal period.” and concluded the paragraph with “Moreover, during pregnancy specific types of worries with respect to labour and the child often occur and are not addressed in commonly used self-report instruments. Therefore they may be overlooked and underreported when assessed with these instruments. [5]."

Writing should be reviewed, especially about grammar and syntax.

We have made every effort to correct grammatical and syntax errors.

What would "mama kits" be?

We now described the MamaKits study in more detail as follows: “In short the study included self-referred pregnant women expressing interest in an internet intervention to treat their depressive or anxiety symptoms. Inclusion criteria were age above 18yrs, being less than 30 weeks pregnant, having depressive symptoms above threshold (i.e. CES-D >16) or anxiety symptoms above threshold (i.e. HADS-A >8) or both of them. One single exclusion criterion was being suicidal. Participants were randomly allocated to receive either a guided internet intervention or care as usual”

The numbers of participants are different in terms of methods and results.

We changed  the text and made clear that numbers in both sections are equivalent.

Method section:  "A total of 349 women expressed interest in the study of which 91 withdrew their participation for several reasons such as feeling better already.  A further 99 were excluded because they did not fulfill the inclusion criteria such as not reaching the symptom threshold.

A total of 159 women completed one measurement at inclusion (T0) and met the inclusion criteria. About 10 weeks after randomization 119 of them also filled out the second measurement (T1)."

General result section:

“A total of 159 women, who completed T0 and met the inclusion criteria enrolled of which 74.8 % (n=119) completed T1.”

What were the inclusion and exclusion criteria?

We added the inclusion and exclusion criteria to the method section as follows:” Inclusion criteria were age above 18yrs, being less than 30 weeks pregnant, having depressive symptoms above threshold (i.e. CES-D >16) or anxiety symptoms above threshold (i.e. HADS-A >8) or both of them. One single exclusion criterion was being suicidal.”

I believe that this review is necessary for the reader to fully understand the findings and discussion. We agree with the reviewer and changed the manuscript accordingly.

Results:  This paragraph could be in the method (participants).

"A total of 349 women expressed interest in the study of which 91 women withdrew 214 their participation for several reasons like feeling better already and 99 were excluded 215 because of not fulfilling the inclusion criteria like not reaching the symptom threshold. A 216 total of 159 women, who completed T0 enrolled and 74.8 % (n=119) completed T1."

We made some changes to the paragraph and moved it to the methods section.

We hope that we have addressed the comments  in such a way that our manuscript is now suitable for publication in IJERPH . We are looking forward to your decision and remain available to answer any further questions.

Sincerely yours, on behalf of all co-authors,

Hanna Heller,

Reviewer 3 Report

First of all I wish thanks the authors for this work.

Below some comments and suggestion:

- line 22: I don't find so proper to use the word 'common problem', but I suggest, as also stated in the abstract to use another form;

- line 22: Please specify if you refer to "Depression and anxiety" or "Depression and anxiety symptoms";

 - line 118: please state whether, in addition to the three instruments you reported, you collected other data and whether an informed consent was signed by the participant. I know that you wrote it in the statement, but please report it here;

- line 188/189: correct DSM5 in DSM-5;

- Table 1: It is somewhat confusing to report the number of subjects 159 at the beginning and then find 119 just below. It would be useful to have the scores also compared between the same subjects at the two times (T0=119 and T1=119);

- Table 1: please write the type of pregnancy (singleton, twins....) and also if some of the women come from in vitro techniques;

- Please format all the tables in the correct way;

- Please highlight the possible clinical implications of your study.

Round 2

Reviewer 2 Report

I congratulate the authors for the article. I have no further comments or suggestions.

Author Response

We thank the reviewer for her or his approval and the congratulations.